# Green LD (BOF) Steelmaking—Reduced CO$_2$ Emissions via Increased Scrap Rate

Bernhard Voraberger *, Gerald Wimmer, Uxia Dieguez Salgado, Erich Wimmer, Krzysztof Pastucha and Alexander Fleischanderl

Primetals Technologies Austria, 4032 Linz, Austria; gerald.wimmer@primetals.com (G.W.);
uxia.dieguez-salgado@primetals.com (U.D.S.); erich.wimmer@primetals.com (E.W.);
krzysztof.pastucha@primetals.com (K.P.); alexander.fleischanderl@primetals.com (A.F.)
* Correspondence: bernhard.voraberger@primetals.com; Tel.: +43-73265924087

**Abstract:** The basic oxygen furnace (BOF) is the dominating primary steelmaking process. It is an autothermal process where hot metal and scrap are used as charging materials. The decarbonization and transformation of integrated BOF steelmaking will be the most important challenge in the coming years. Steel scrap is a charge material without new CO$_2$ emissions, whose availability is expected to grow significantly and will play a key role in this decarbonization process. Several solutions have been developed by Primetals Technologies to provide additional energy for processing higher scrap rates in integrated BOF steelmaking. Such solutions include simple upgrade packages installed on existing converters such as process models for heat optimization, post-combustion, and scrap preheating lances. For higher scrap rates from 30% to 50%, a combination blowing converter and JET converter is required to provide sufficient mixing during scrap melting and the highest heat transfer from the increased post-combustion. Hybrid EAF–BOF operation and limitations regarding scrap quality also need to be considered for the transformation of steelmaking. Scrap sorting and processing can be a solution to reduce residual levels in crude steel for high scrap rates. Based on reference plant data, the CO$_2$ reduction potential of the presented solution versus the effort and complexity of implementation is compared.

**Keywords:** BOF; scrap rate; CO$_2$ emissions; decarbonization; design scrap; process optimization; post-combustion lance; scrap preheating; KOBM; JET converter; EAF; PREMELT process

## 1. Introduction

The global steel industry is responsible for nearly 20% of global coal consumption, which is mainly used for the reduction in iron ore in the blast furnace (BF) process to generate hot metal. This coal-based reduction process in the BF and the following oxygen steelmaking process leads to specific CO$_2$ emissions of approximately 2 tons per ton of steel. With a current global production of about 1850 MTPA, 70% via the BF–BOF (basic oxygen furnace) route [1], the steel industry is the largest industrial CO$_2$ emitter and accounts for up to 9% of worldwide carbon dioxide emissions. International agreement to achieve net-zero before 2050 for the majority of the steel producing countries has been made. With increasing pressure from governments to reduce CO$_2$ emissions to fight climate change and ever-increasing emissions trading systems (ETS) and carbon taxes, many steelmakers have announced decarbonization pathways for their steel products [2]. In Figure 1 below, on the left side, the specific CO$_2$ emissions per ton of output product for the typical raw materials such as hot metal, direct reduced iron (DRI), and scrap are shown.

It can be seen that hot metal has by far the largest "backpack" of CO$_2$ emissions, with around 1.5 tons of CO$_2$ per ton of liquid steel. Direct reduced iron with natural gas has less than half the emission of hot metal via the BF process. Future hydrogen-based direct reduction is already becoming close to zero, where only some small amount of

scope 2 emissions for operating the plant will remain. The only raw material for steelmaking with zero $CO_2$ emissions is steel scrap. Therefore, an increased usage of steel scrap, which can be recycled infinite times, will play an important role in the decarbonization of the steel industry alongside direct reduction-based steelmaking. $CO_2$ emissions are grouped into three scopes: scope 1 includes direct emissions, e.g., combustion processes, scope 2 includes indirect emissions, e.g., from electricity, and scope 3 includes other indirect emissions from sources not attributed to the plant, e.g., raw materials [3].

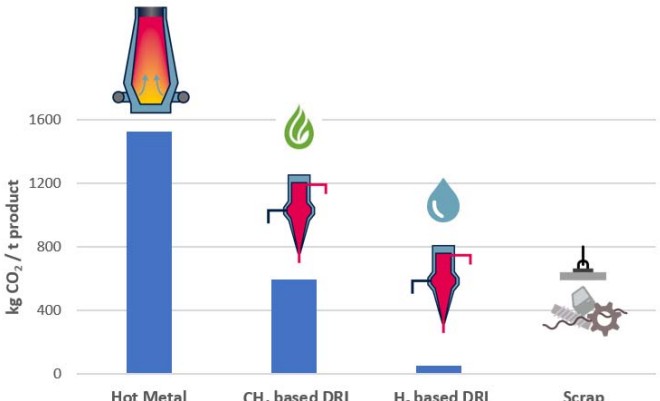

**Figure 1.** Scope 1 and scope 2 carbon dioxide emission per ton of output product (e.g. per ton of hot metal) for typical steelmaking raw materials (grid factor of 0.45 kg $CO_2$/kWh considered).

Increasing global scrap usage for steelmaking shows that all available scrap is currently in use, and this will not change in the future when scrap availability is further increased. Forecasts predict scrap availability from 900 MTPA [4] to 1300 MTPA [5] in 2050 (see Figure 2). Since global steel production is expected to grow far less than scrap availability, forecasts show around 2200 MTPA in 2050 [1,6]. The higher scrap availability will lead to new scrap-based electric arc furnace (EAF) steelmaking plants and increased scrap utilization for integrated steelmaking. Processing higher scrap rates, e.g., in the BOF, is an easy and fast measure to reduce the carbon footprint of the integrated plant and therefore will become more popular. Unfortunately, scrap availability and the residual limitation for high-quality steel grades will not allow a switch to entirely scrap-based steelmaking. Therefore, virgin material such as iron ore will remain the dominating raw material long term.

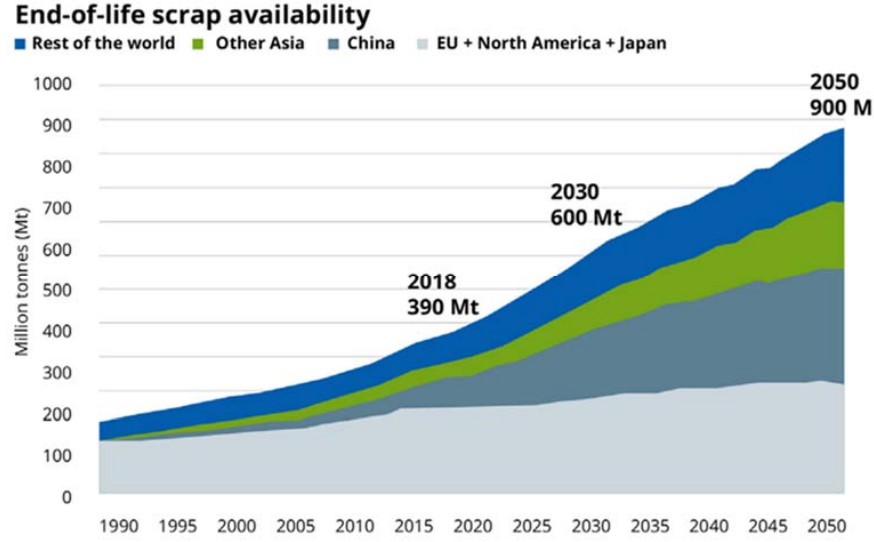

**Figure 2.** End of life scrap availability in million tons per year (MTPA). Reprinted with a permission from ref. [4].

In this paper, the challenges and limitations of processing higher scrap rates will be discussed, which is followed by solutions for scrap processing, higher scrap rates in integrated BOF steel plants, and a hybrid EAF–BOF steelmaking operation as part of the transition toward net-zero carbon emissions. Finally, the $CO_2$ reduction potential of the presented solutions is compared and set into relation to the complexity of implementation.

## 2. Challenges for Processing Higher Scrap Rates

A general topic of scrap-based steelmaking is the contamination of scrap with dust, dirt, and residual elements. Such residual elements are non-ferrous metals such as copper (Cu), tin (Sn), chromium (Cr), molybdenum (Mo), and other harmful elements such as sulfur (S) and phosphorus (P). Some of them, such as copper, are nobler than iron and cannot be removed by oxidation in the steelmaking process, limiting the extent to which they can be allowed in the scrap. Figure 3a below shows the residual levels for different types of scrap. These levels are higher than the maximum allowed amount for certain steel qualities indicated by the orange lines. Such limitations mainly affect the scrap-based EAF steelmaking process, but this topic is also relevant for BOF steelmaking with increased scrap rates.

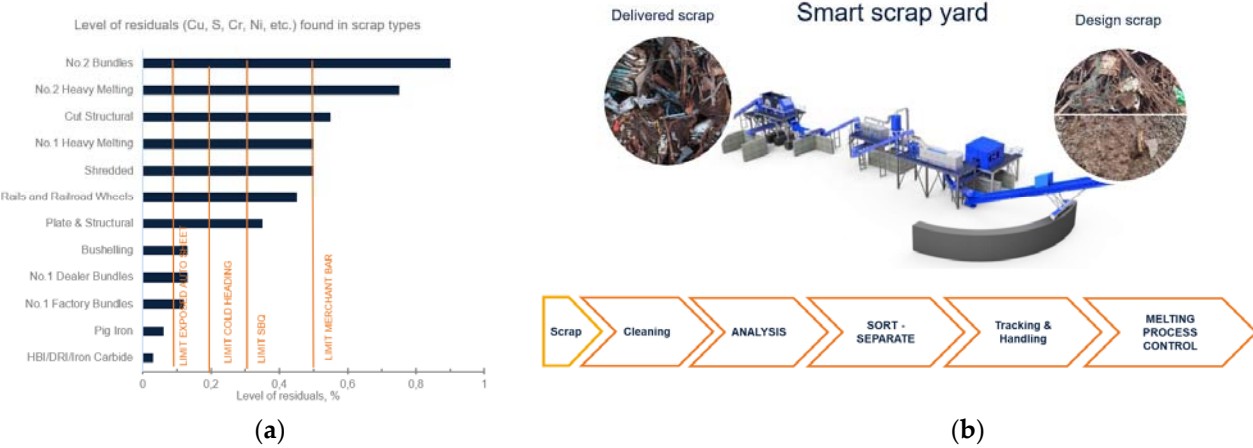

(**a**)        (**b**)

**Figure 3.** (**a**) Residual element level for different scrap types and limits for steel groups (**b**) Overview scrap processing plant and different process steps.

Global scrap usage increased in the last few years, especially in China, where the scrap usage since 2017 has more than doubled. The main scrap exporters worldwide are the EU and the USA. The largest scrap importers are Turkey and India [7]. More than 20 MTPA of predominantly low-quality scrap is exported by the EU each year, and this has increased by 20% since 2011. The decarbonization of the steel industry and the resulting processing of higher scrap rates will lead to increased competition for high-quality scrap with lower residual amounts. Hence, in the future, exports of lower quality scrap might be reduced as technologies for cleaning, sizing, analyzing, and sorting of scrap emerge to produce so-called "design scrap". With such technologies, low-quality scrap can be sorted and processed into different types of scrap, and the parts that fulfill the specifications for the desired steel grades can be used. In addition, diluting low-quality scrap with DRI and pig iron or hot metal as a flexible charge mix will be applied to achieve the desired steel grade. Figure 3b above shows a smart scrap cleaning plant. Features such as image processing with AI support will allow scrap classification of the incoming material. Unwanted parts such as electric motors or safety-critical parts such as pressure vessels can be removed in this step. Further sizing scrap with shredders followed by chemical analysis will allow separation of the resulting design scrap according to its composition. Smart tracking and handling solutions combined with advanced process automation to calculate the optimum charge mix according to final steel composition requirements will produce high-quality steel with higher scrap rates.

Yet, scrap quality and availability are not the only challenges. Logistics and the melting process in the BOF show limitations. Regarding logistics, a limitation for implementing a higher scrap rate might be the volume of the scrap chute. Using a second scrap chute requires extra time for charging at the BOF and will impact process times, crane utilization, and complete scrap logistics. Hence, a complete scrap logistics and preferable production plan need to be evaluated with digital logistic simulation tools to identify bottlenecks and find the best solutions for implementing a higher scrap rate at the desired production capacity.

The limitations regarding the BOF steelmaking process with a higher scrap rate, such as the energy balance and required mixing power to melt all scrap pieces in time, will be discussed in the following section.

## 3. BOF Steelmaking Process: Energy and $CO_2$ Balance

Currently, around 70% of global steel production is based on the BF–BOF route. The remaining 30% is based mainly on scrap-based EAF steelmaking and a smaller but increasing share on DRI–EAF-based steelmaking [5,6]. Unlike the electric arc furnace, which uses electric energy as the main source to melt down the charged materials, the BOF process is an autothermal process that does not require external energy input. The main energy input for the LD (BOF) process comes from the sensible heat of the hot metal and chemical reactions such as the oxidation of carbon (C), silicon (Si), and a small amount from the post-combustion of carbon monoxide (CO). While Si is combusted in one step to silicon dioxide ($SiO_2$), C is combusted in a first step in the steel bath to CO only and during a second step outside the bath to $CO_2$. The latter step is called post-combustion and generates about two-thirds of the total energy released if carbon is combusted to carbon dioxide ($CO_2$).

In conventional BOF operation, the post-combustion degree is rather low, only 8–12%. Hence, only a small part of the total energy is used inside the converter. A gas with a higher calorific value is leaving the process, typically collected in a gas holder, and used for heating purposes in the plant later. This process is called BOF gas recovery.

Scrap and other solid charges such as hot-briquetted iron (HBI), DRI, sinter, or iron ore are added as cooling the material to the BOF to equalize the heat balance and compensate for excessive heat from the hot metal. Lime, limestone, and burnt or raw dolomite are added as slag formers. Figure 4a below shows the energy balance of a BOF converter for a scrap rate of 20% in a simplified manner. The main energy input is around 60%, and the sensible heat of hot metal is typically about 1350 °C, which is followed by the combustion of carbon to CO at approximately 18%. All chemical reactions in the liquid steel and slag account for 34%, and the post-combustion of CO to $CO_2$ in the BOF process gas provides around 5% of the energy input, assuming a standard post-combustion degree. On the output side, the main energy is the sensible heat of steel at around 1670 °C, consisting of the sensible heat of steel coming from the hot metal with around 56% and the energy of the scrap melting with around 20%. The sensible heat of slag with 12.8%, the sensible heat of the process gas with 8.5%, and the heat losses with about 3% are concluding the energy output. In Figure 4b, the energy balance for 20% scrap rate is compared with a higher scrap rate of 25%. For a higher scrap rate, the sensible heat of hot metal and heat from the chemical reaction decrease with the lower amount of charged hot metal. In addition, the post-combustion decreases because of the lower amount of CO generated. With the higher scrap rate of 25%, additional energy is required for melting, assuming the same chemical composition and temperature of hot metal and the same tapping conditions. The additional energy required for melting more scrap with less hot metal is shown in red below. The total energy output of steel is the same for the higher scrap rate, but less comes from the hot metal. With a lower hot metal ratio, it is assumed that slag and off-gas decrease. Heat loss will stay relatively constant since the main factors influencing heat loss—the charging and tapping temperatures, refractory, insulation, and the processing time—are not changed. Heat loss is understood as the heat transfer and radiation from the vessel to

its surroundings. Since slag and off-gas amount is slightly lower with a higher scrap rate, a similar trend is expected for the total energy compared to the 20% scrap rate.

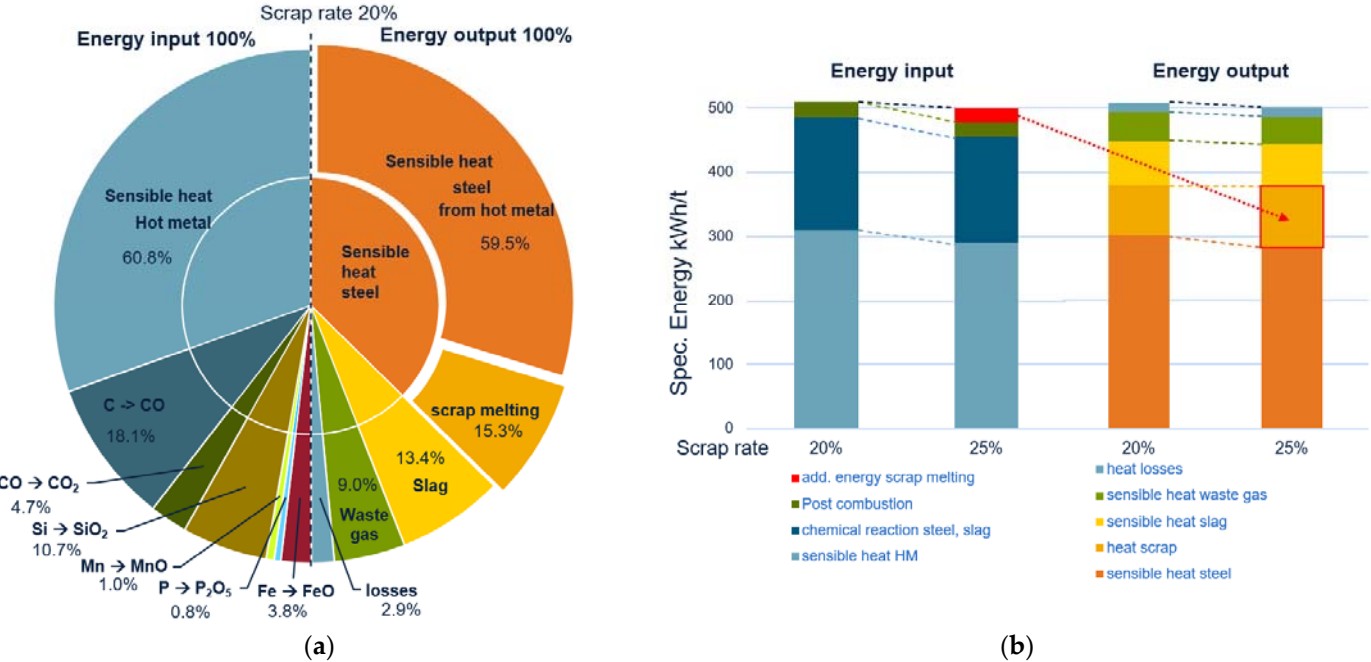

**Figure 4.** (**a**) BOF energy balance for 20% scrap rate (**b**) comparison of energy input and output for 20% and 25% scrap rate.

Since the amount of combustibles in the BOF process is limited and heating agents such as ferrosilicon or coal show limitations regarding process and economics, the share of solid charges such as scrap and HBI/DRI is limited. Depending on the hot metal composition, temperature, plant-specific operation conditions, and corresponding heat losses, the BOF process is well balanced at the scrap rate of 15 to 20%. The possible solid charge rate decreases with the increasing share of HBI and DRI, since these materials have a higher cooling effect than scrap because HBI and DRI contain some percentage of iron oxide that was not reduced in the reduction plant as well as some non-metallic fraction called gangue, which requires more energy for melting. Hence, the maximum share of scrap that can be processed in the BOF is higher than the maximum share of HBI. In the BOF, around 10% of the charge mix can be HBI/DRI, which is either charged with the scrap chute and/or with the material handling system. Furthermore, HBI/DRI have higher $CO_2$ emissions compared to scrap, are more difficult to melt, and special care for the charging step and mixing during processing is required to avoid slopping. Therefore, in the following chapter, the focus will be on scrap as a solid charge.

Another point that restricts the scrap rate in conventional BOF converter steelmaking is bath mixing. Strong mixing of the bath is required to keep temperature gradients small and avoid cold spots to melt down high amounts of scrap per heat quickly. Hence, proper bottom stirring is necessary throughout the vessel lining campaign [8].

$CO_2$ balance of integrated BF-BOF steelmaking and specific $CO_2$ emissions per ton of output product with a scrap rate of 16%, including credits for gas and heat recovery from off-gas and granulated blast furnace slag (GBFS), is shown in Figure 5 below. The scope 1 emissions from ironmaking and here especially from the blast furnace are dominant along the process chain. At the BOF, the specific scope 1 emissions from the decarburization process are lower by far, with around 160 kg $CO_2$ per ton of liquid steel, and scope 2 emissions for electric power, oxygen generation, and dry primary dedusting system in total are 35 kg $CO_2$ per ton of liquid steel. In green, the $CO_2$ credits are shown in Figure 5. In the case of a dedusting system with suppressed combustion, BOF off-gas has a good heating value and is typically collected in a gasholder for further usage. Depending

on the BOF operation with such gas recovery, $CO_2$ credits of 30–40 kg per ton of liquid steel can be achieved. Additional credits are possible with heat recovery of BOF off-gas to produce steam and substitute other fossil fuels such as natural gas, which are in a range of 10–30 kg per ton liquid steel. Newly developed solutions for by-product recycling of slags and dust on the BOF, which allows the recovery and reuse of metal and mineral fractions, have the potential for additional credits.

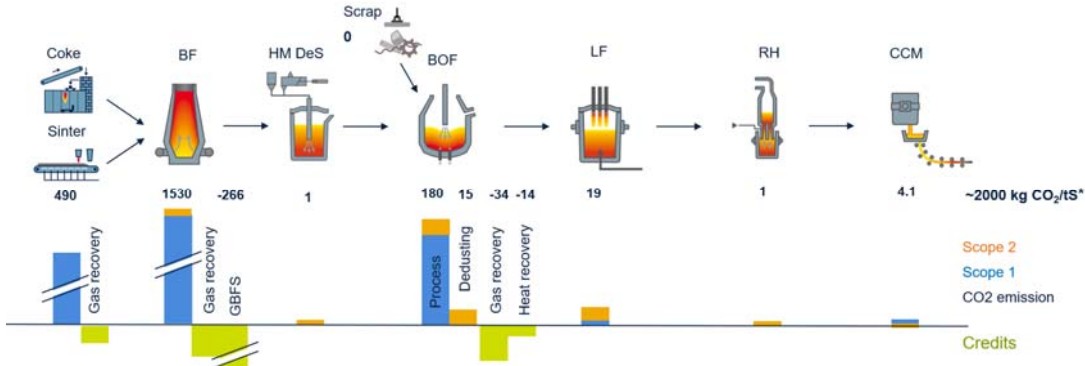

**Figure 5.** Scope 1 and scope 2 $CO_2$ emissions for integrated BF-BOF steelmaking including credits per ton of output product. * Specific values based on an electricity grid factor of 450 g$CO_2$/kWh.

## 4. Solutions for Increased Scrap Rate for BOF Steelmaking

From the $CO_2$ balance, it can be seen that increasing the scrap rate will reduce the hot metal amount and therefore show promising potential to reduce $CO_2$ emissions of the integrated steelmaking route while in parallel allowing for higher raw material flexibility. Several measures exist to increase the energy input and decrease the energy output at the BOF to allow for operation with higher scrap rates. The next section will discuss efforts such as heat optimization, scrap preheating and premelting, higher post-combustion, combined blowing, and the JET converter.

### 4.1. Heat and Process Optimization Combined with Level 2 Process Models

Efficient tools for optimizing the heat balance and the BOF process control are advanced process automation tools such as Level 2 (L2) process models and online heat scheduling. Primetals Technologies L2 Process Model allows fast and accurate pre-calculation and simulation of the heat with the available raw materials, process data, and steel grade targets. Combined with off-gas analysis, an automatic blow stop and high accuracy for achieving carbon targets and temperature can be guaranteed. A sublance measurement will increase accuracy further. Thus, reblows can be avoided, the usage of scrap as coolant maximized, and other cooling additions such as iron ore avoided. The process models allow for the maximum possible scrap rate and increased raw material flexibility. (For the interface of an L2 Process Model, see Figure 6b.)

For example, with higher silicon content in hot metal or with higher hot metal temperature, the scrap rate can be increased very flexibly depending on hot metal conditions. Possible cooling additions such as iron ore or sinter, regularly used in the case of hot metal variations, can be replaced by scrap; see Figure 6a. Up to a 9% scrap rate increase and similar reduction in BF–BOF $CO_2$ emissions can be achieved with heat optimization and advanced L2 process automation for BOF plants operating at very low scrap rates. The increase in scrap rate that can be achieved with such measures always depends on the starting point and is, of course, smaller for a plant that is already well optimized. For plants operating at around 20% scrap rate, a further increase will be possible only in a moderate way up to 3%, including heating agents. The amount of heating agents such as ferrosilicon, which can be added, is limited due to process and economics. Therefore, it is typically used for correction in daily operation with a minor scrap rate increase of 1–2% only. Exact figures need to be analyzed depending on plant-specific conditions.

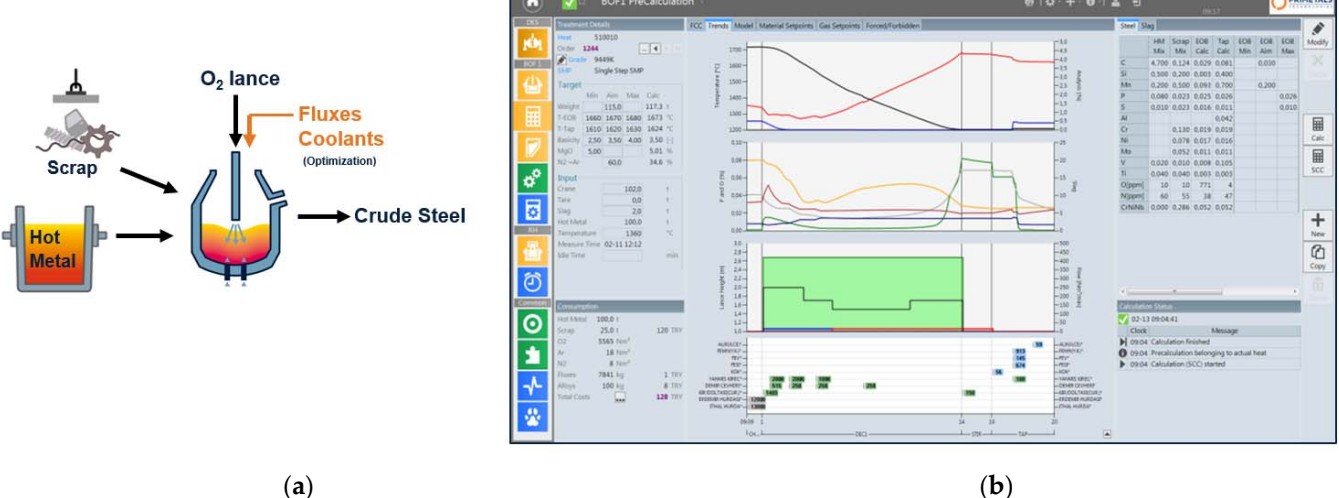

(**a**)                                                                                       (**b**)

**Figure 6.** (**a**) Heat optimization through replacement of cooling additions with scrap. (**b**) Interface of Level 2 process model for process simulation and control. The calculated temperature (red), carbon (black), and silicon content (red) are shown in the middle-upper diagram. Slag composition, blowing pattern, and material addition schema are illustrated in the lower diagrams.

Lowering the energy output can be achieved by reducing transport times and improving logistics. Connecting the process automation of the continuous caster machine with previous secondary and primary steelmaking is what the online heat scheduling tool does to improve the logistics and, as a result, also reduces heat losses. The caster scheduler generates heat sequences in line with melt shop capabilities. The Online Heat Scheduler (OHS) plans the optimal route through the melt shop based on desired casting sequences in a pull principle. It reacts immediately to possible delays in the steel plant and coordinates hot metal demand and delivery. The OHS allows for reduced waiting times in the steel plant logistics and minimizes heat loss.

*4.2. Scrap Preheating with Scrap Preheating Lance*

For integrated plants where the BOF converter is not the bottleneck and plants with a lack of hot metal or high variation in hot metal supply, scrap preheating represents a good opportunity to increase the scrap rate and raw material flexibility. Several scrap preheating options inside and outside the BOF have been investigated. Performing preheating of the scrap inside the vessel is recommended to avoid any handling of the preheated scrap, which would cause a lot of dust generation in the bay and avoid heat loss during the transport of the preheated scrap. The utilization of heat from BOF off-gas would be perfect for scrap preheating. Still, with limited space availability, resulting in a high falling height of scrap, and short cycle times, this solution seems not very feasible. Currently, the most efficient solution for scrap preheating is a burner lance installed instead of one of the top oxygen lances on the second lance carrier, which is used for preheating after scrap charging. The remaining oxygen top blowing lance is upgraded with a quick coupling system to ensure fast exchange and high availability of blowing lance. The lifetime of preheating lance will be much longer than the typical oxygen-blowing lance. No wear is expected on the burner tip, since solid scrap preheating will generate less dust and no skull formation, and therefore, operation of at least one lining champaign can be expected. (For a picture of a scrap preheating (SPH) lance and lance tip with a burner head, see Figure 7a below.)

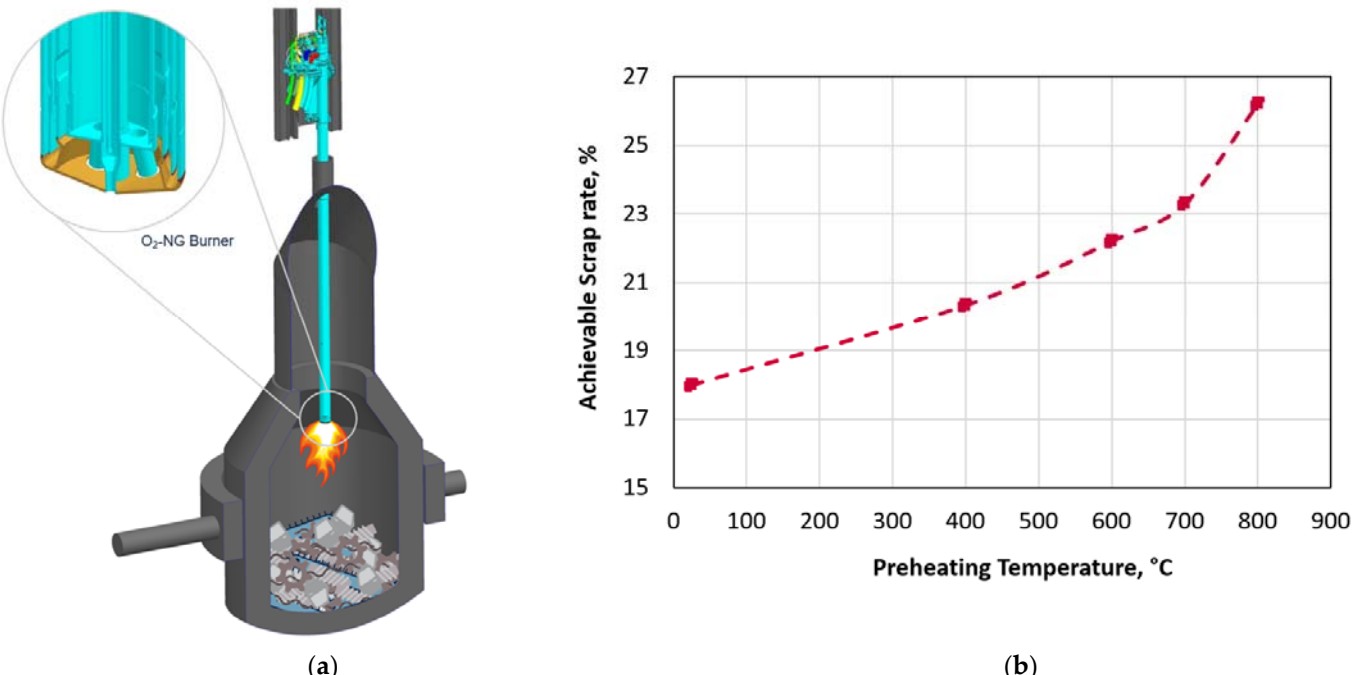

<div align="center">(<b>a</b>)</div> <div align="center">(<b>b</b>)</div>

**Figure 7.** (**a**) O₂ Natural Gas Burner used as scrap preheating lance inside the BOF. (**b**) Influence of scrap preheating temperature on scrap rate at defined burner power.

The energy transfer from the burner flame to the scrap takes time. It largely depends on the scrap size and density—the longer the preheating time, the higher the temperature of the scrap and the maximum possible scrap ratio. The preheating temperature is limited by the burner power, heat transfer, preheating time, and iron oxidation in the scrap. Therefore, typical preheating temperatures are up to 800 °C to avoid too high iron oxidation and less efficiency. Oxygen natural gas burner power can vary from 10 to 35 MW depending on the converter size and preheating time. For a test case of a 300 t BOF converter, a 35 MW burner has been developed. The power of the burner can be adjusted to be flexible in usage for the preheating of scrap, the heating up of the refractory, or off-gas. The burner is equipped with an ignition system and flame control monitoring for safe operation.

Preheating time needs to be between 5 and 15 min and will increase the scrap rate by up to 8%. Scrap increase depends on the preheating temperature (see Figure 7b above), which is influenced by several factors such as the burner power and converter size. The slope of the curve is influence by the change in specific heat capacity due to the solid phase transformation of iron at around 700 °C.

The additional time required for preheating the scrap and charging a higher scrap amount can only be partly compensated for by reducing the charging time for hot metal and reducing the main blowing time due to the reduced hot metal ratio. In total, the tap-to-tap time will increase if scrap preheating is applied. For plants where the BOF converter is not the bottleneck, the scrap-preheating lance can be a key element to increase the scrap rate. Figure 8 shows typical process times for BOF operation with the scrap-preheating lance.

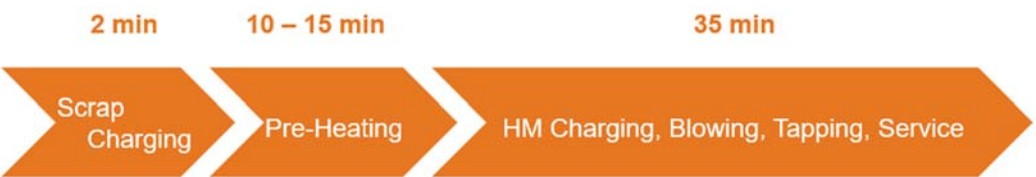

**Figure 8.** Typical processing time for BOF with scrap preheating.

Further increase in the scrap rate is possible with scrap premelting in an electric arc furnace or induction furnace (see the PREMELT process in Section 6).

### 4.3. Dual Flow Post-Combustion Lance

Another measure is to use the energy from the post-combustion of BOF gas to increase the energy input of the liquid steel bath. The typical BOF process gas consists of approximately 10% $CO_2$ and 90% CO. Through post-combustion of CO to $CO_2$, double the amount of energy is realized compared to the combustion of C to CO. A post-combustion lance is installed instead of the normal top blowing oxygen lance for this purpose (see Figure 9a). Such a lance has a similar lance tip as a normal oxygen blowing lance—called the main port—and a secondary port above the main port where additional oxygen is blown in. The secondary port is designed to ensure that the oxygen blown through this port is mainly used for post-combustion of CO coming from the bath. The nozzle arrangement, design, and flow rates have been optimized with comprehensive computational fluid dynamics (CFD) simulation (see Figure 9a) to ensure that the heat generated in post-combustion is transferred by in large to the liquid bath and not to the off-gas, the lance body, or the refractory. The generation of CO changes during the blowing process. A high CO formation can be observed during the main decarburization period, whereas early and late decarburization show lower CO formation.

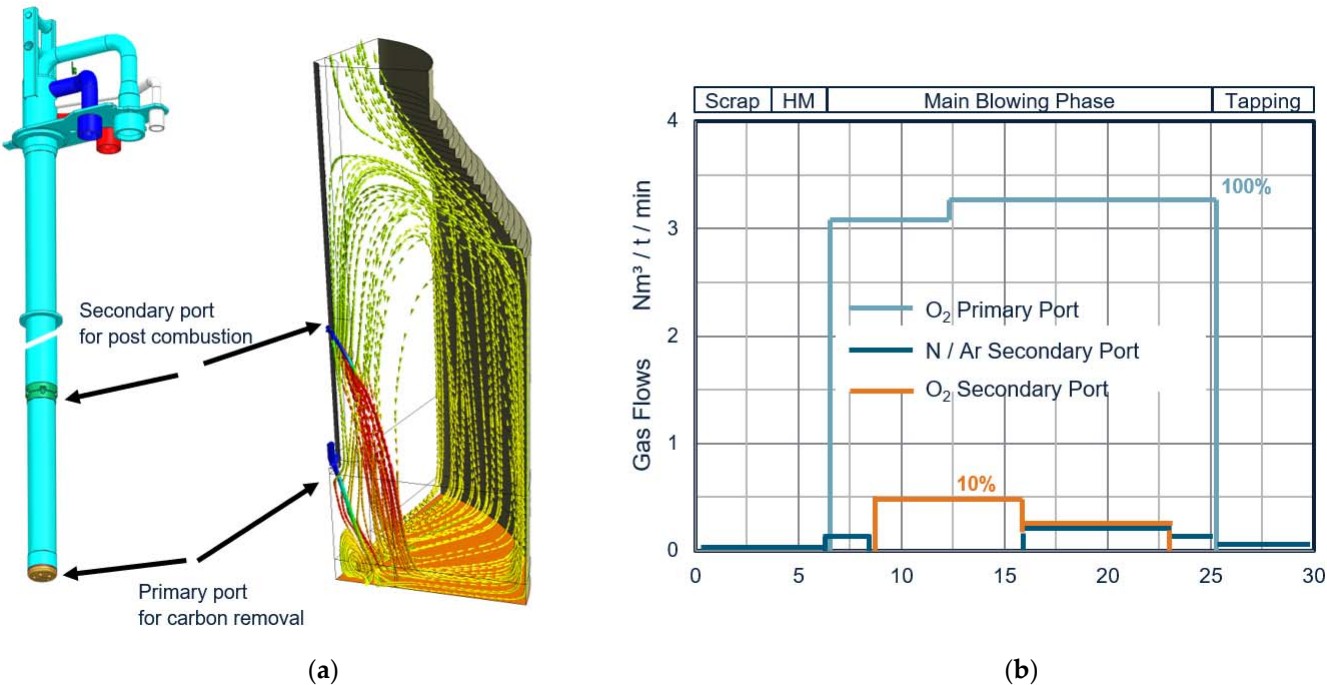

(**a**)          (**b**)

**Figure 9.** (**a**) DFPC lance and CFD simulation for optimized heat transfer depending on BOF geometry (**b**) blowing pattern and specific flow rates for application of DFPC lance.

The flow at the secondary port needs to be controlled independently from the flow at the main port to control post-combustion. Hence, a second control line needs to be installed, and the lance is called a dual flow post-combustion (DFPC) lance. In addition, mixing with nitrogen (or argon) is installed for the oxygen blown through the secondary port providing a second level of control of post-combustion degree and for cooling of the nozzle and to avoid blocking. Blowing patterns for the DFPC lance show that the lance allows for an increase in the post-combustion rate from around 10% for a normal BOF up to 17% during the main decarburization period with high heat transfer efficiency increasing the scrap rate by approximately 4%. An exact figure depends on plant conditions such as hot metal and scrap composition.

There is an essential difference of the above-mentioned DFPC lance from a standard single flow post-combustion lance with post-combustion ports where only pure oxygen ($O_2$) is applied, as post-combustion gas and the oxygen flow for the post-combustion is fixed and not separately controlled. Such single flow post-combustion lances are very

common to reduce skull formation on the blowing lance and converter mouth and top cone. The fixed $O_2$ flow rate for single-flow post-combustion lances is around 5% of the total oxygen flow rate. Due to this small $O_2$ amount, the small flow rate, and the different nozzle arrangement of the secondary ports for such lances, no impact on the scrap rate could be observed. Whereas reference and test installation of DFPC lances have shown that scrap rate increase is possible and further advantages such as the reduced skulling of the BOF mouth and top cone as well as the lance below secondary ports and better dephosphorization due to better mixing of slag could be achieved. The $O_2$ flow rate on the main lance tip at the DFPC lance is not changed compared to the standard blowing lance, and therefore, the total blowing time remains unchanged. Hence, the same high productivity can be ensured.

Consider that all the above-described areas will allow scrap rates between 25% and 30%. For such high scrap rates, scrap logistics need to be checked in detail, especially the capacity of the scrap chute. An enlargement of scrap chute and increase in scrap density by scrap processing are measures to improve. If this is not sufficient, scrap needs to be charged with two chutes which, of course, will take additional time. Furthermore, good bottom stirring is required to ensure intensive mixing of the bath and proper melting of all the scrap added. The last critical point is the ignition at the blowing start, which is not that easy in the case of high scrap rates and the cold surface temperatures caused by it.

A change from the standard BOF converter with oxygen top blowing and inert gas bottom stirring to the combined blowing converter with oxygen blowing from the top and bottom can be completed to overcome the limitations for processing higher scrap rates.

*4.4. Combined Blowing and JET Converter*

Stable processing of a high scrap rate is achieved by changing from normal BOF operation to combined blowing converter, known as KOBM, or Klöckner Oxygen Blown Maxhütte. The oxygen blown from the bottom via gas-cooled tuyeres leads to intensive mixing of the bath. This intensive mixing ensures uniform scrap melting and allows for softer blowing with the top lance, which automatically results in a higher post-combustion rate. In addition, the tuyeres installed in the converter bottom can be used for scrap preheating. Such a combined blowing converter is, for example, operated at AM Dofasco, Canada, and an increase in post-combustion and scrap rate has been reported.

Primetals Technologies has several references for such KOBM converter, and the most recent reference of KOBM, which is currently under construction in China, shows the typical advantage of KOBM technology compared to BOF, such as the following:

- Faster slag formation and less slopping due to the lime injection, good mixing, and process close to equilibrium—less material loss, less cleaning effort, less maintenance effort
- Less iron oxide (FeO) in the slag due to better mixing (FeO in the slag partially reduced again by C from the bath) and therefore higher yield,
- Higher post-combustion degree due to increased lance height (more soft blowing).

In Figure 10b, below, the percentage of iron in slag at a specific tapping composition is compared for standard BOF (LD) with bottom stirring, KOBM, and OBM (Oxygen bodenblasen Maxhütte or "Oxygen Bottom-Blowing Furnace"). It clearly shows that the bottom blowing converter has an advantage concerning yield and iron losses due to the very efficient homogenization.

Other advantages of KOBM, such as the shorter process time, better phosphorus removal, and less dust formation are in evaluation and will be reported after completion.

Approximately 30% of the total oxygen is blown through the bottom, accounting for 250 $Nm^3$ per minute (see Figure 10a above). The bottom lime injection rate is 700–900 kg/min. The scrap rate is relatively low at this converter, with approximately 10%, since the main target is to maximize iron ore additions and raw material flexibility.

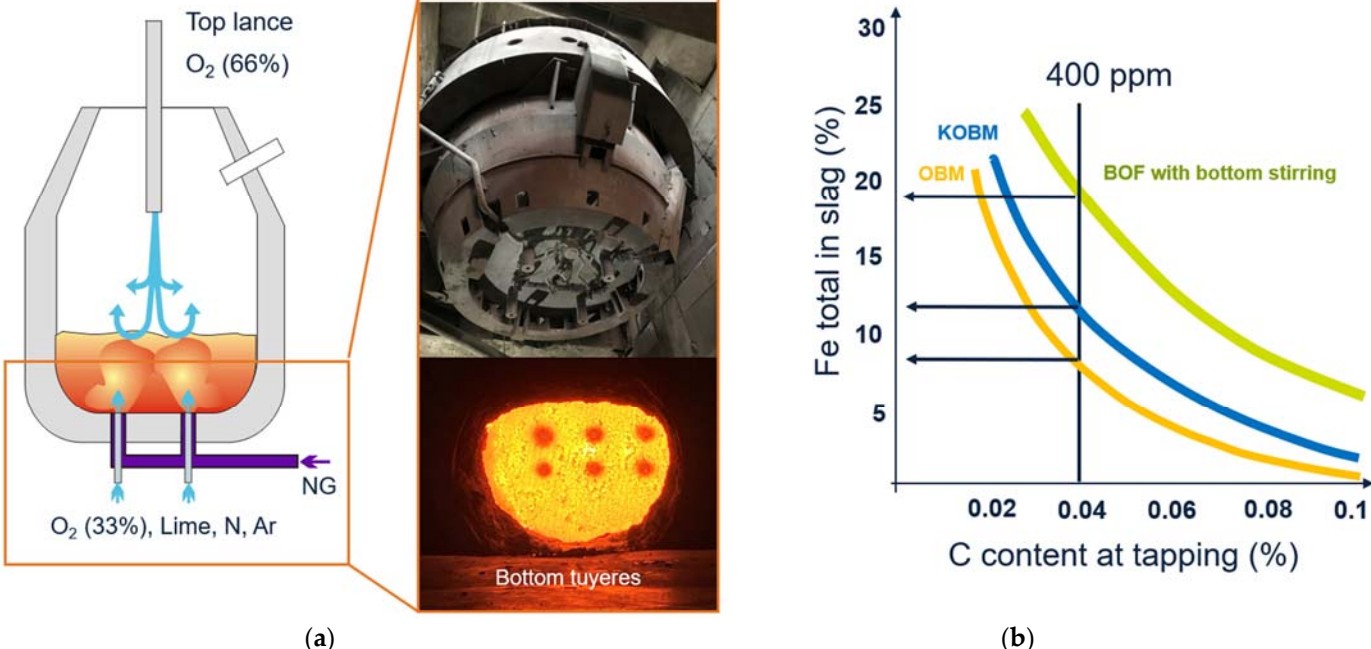

**Figure 10.** (**a**) KOBM converter with top and bottom blowing system and lime injection. (**b**) Total iron content in slag at certain carbon content at tapping for different converter processes: BOF process with top blowing and bottom stirring, KOBM with combined blowing, and OBM with oxygen bottom blowing only.

There are also challenges of KOBM operation compared to the BOF, such as the higher maintenance effort for intermediate hot bottom exchange and tuyeres operation. Due to the extensive bath mixing of the bottom and combined blowing converter, vibrations can be observed, which requires stiffer mechanical design and, in some cases, a vibration-damping system.

The JET process was developed to further increase the scrap rate by higher post-combustion degree and the addition of heating agents by the injection of coal directly into the steel bath. The JET process comprises a bottom blowing converter with coal and lime injection combined with a hot blast lance [8,9]. The coal injected via the converter bottom as well as the carbon already dissolved in the hot metal is again combusted in two steps—the combustion to CO in the steel bath followed by post-combustion to $CO_2$ outside the steel bath.

Efficient mixing is required to ensure the highest post-combustion and heat transfer efficiency (I). This is achieved by a hot blast blown with a lance from the top onto the bath. The hot blast consists of air that is enriched with oxygen to about 30% and heated up in a pebble heater to 1300 °C (see the principle in Figure 11b). Due to the high speed and the high volume of the hot blast, a jet with a very high penetration length is formed, and a lot of surrounding media is sucked into the jet. This leads to an excellent mixing rate inside the converter; the CO coming from the bath is intensively mixed with the oxygen in the hot blast, and combustion to $CO_2$ takes place. The intensive mixing with the hot blast ensures post-combustion up to 60% and a high HTE leading to an off-gas temperature of about 150 °C above the steel bath temperature.

Figure 11a shows the main components of the JET process, the hot blast system with its core component, the pebble heater, and the bottom blowing converter. The pebble heater is an efficient regenerative heat exchanger that uses pebbles to store energy. Due to the high surface area of the pebbles, they have very high storage power and are ideally suited for short-term heat storage. This, in combination with the high storage density of the pebbles, leads to a very compact design of the pebble heater.

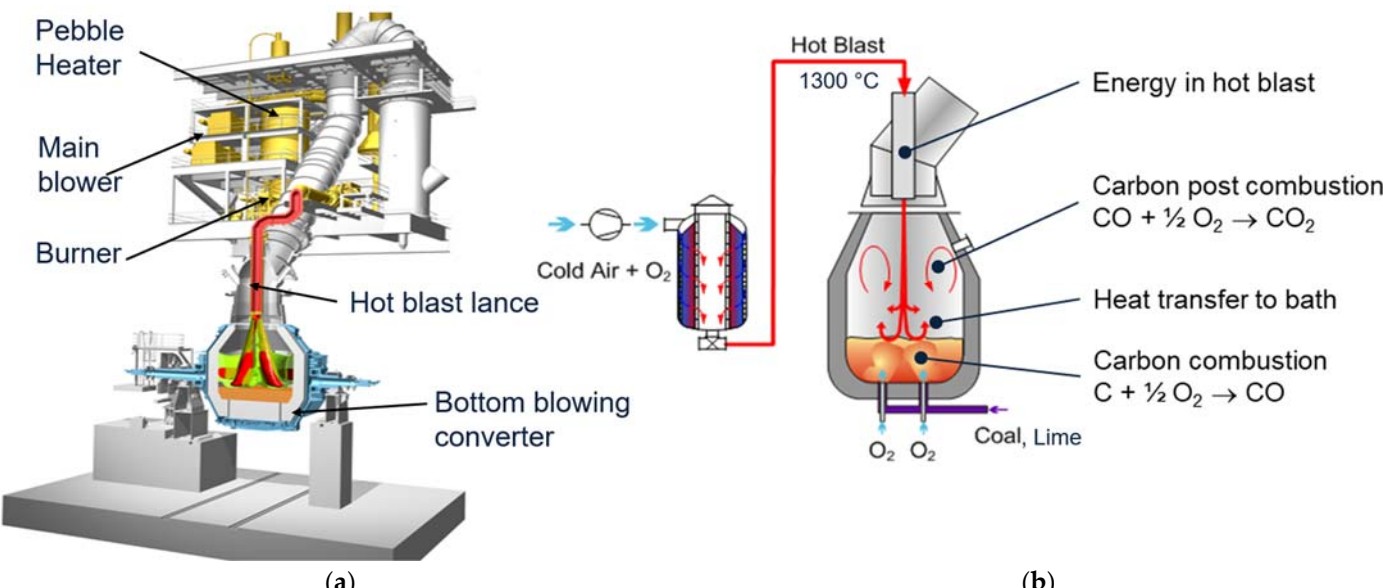

(**a**)                                                                                          (**b**)

**Figure 11.** (**a**) Main components of the JET converter and hot blast system. (**b**) Process and the principle of JET converter: Hot blast system for highest post-combustion and bottom blowing converter with lime and coal injection.

In parallel to the hot blast from the top, oxygen is blown through the converter bottom to decarburize the hot metal. The bottom blowing leads to excellent bath mixing and allows for fast and efficient melting of large scrap amounts.

A full industrial reference for the JET process was implemented in POSCO, Pohang on a 100 t converter (see Figure 12 and [8,9] for details). The principles of the process and performance have been proven in this reference and the design of the equipment. A wide range of scrap rates have been processed successfully, including heats with 50% scrap. In the picture below, a photograph of the converter, the bottom is shown, highlighting the bottom blowing system installed. This system is capable of oxygen blowing as well as coal and lime injection

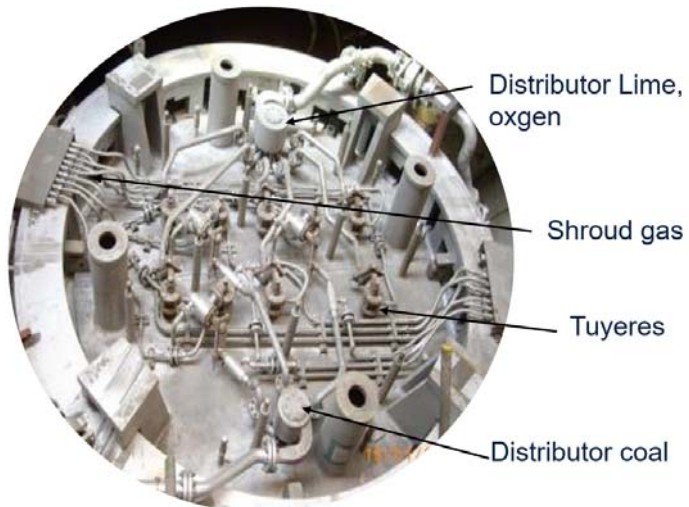

**Figure 12.** Converter bottom of 100 t JET converter at POSCO, Pohang.

## 5. BOF vs. EAF Steelmaking

Off-course direct comparison of BOF and EAF is difficult, since both aggregates are used with different charge mixes. Scrap-based steelmaking is commonly known in the application of electric arc furnaces (EAF). The EAF is a very flexible furnace regarding the

charge mix, and thanks to external energy input provided by the electrodes, up to 100% of solid charge mix (scrap, HBI, DRI) is possible. With a hot metal lauder, hot metal can be charged. The Primetals Technologies flexible furnace that can handle beside scrap and DRI also high ratios of hot metal is called EAF Fusion. For the BOF as an autothermal process, the scrap rate is limited for conventional BOF steelmaking without additional measures at around 20%, optimized operation with solutions described in Section 4 are possible at about 30%, and for special and combined blowing converters, even higher scrap rates are possible. So, the charge mix of BOF and EAF usually is different, and a direct comparison not possible. However, a general comparison of typical BOF and EAF operations is shown in Table 1 below. With over 60% of hot metal in the charge mix, typically, the BOF is the more efficient furnace due to lower heat losses, better decarburization, shorter process time, and lower conversion cost. Below 60% of the hot metal ratio, the EAF will be the preferred solution. For some steel grades with very low nitrogen levels, such as external automotive grades, scrap-based EAF steelmaking will not be feasible, because nitrogen pick up around the electric arc is high, and target values below 20 ppm nitrogen are difficult to be achieved. For EAFs running with high DRI/HBI shares, nitrogen levels of around 30 ppm are possible, [10], but still, values are higher than for BOF. Another difference is the longer refractory lifetime and shorter process time which leads to higher productivity.

**Table 1.** Comparison of typical reference data from BOF and EAF operation.

| KPI | Unit | BOF | KOBM | Scrap EAF | Flexible EAF | DRI EAF |
|---|---|---|---|---|---|---|
| Typical charge mix | |  |  |  |  |  |
| HM/scrap/DRI | % | 80/20/0 | 65/30/5 | 0/100/0 | 40/40/20 | 0/30/70 |
| Yield | % | 90.5 | 91.5 | 89.0 | 89.5 | 87 **** |
| tap to tap time | min | 40 | 40 | 50 | 45 | 55 |
| Nitrogen level EOT * | ppm | 16–50 | 16–50 | 60–100 | 35–80 | 25–60 |
| Refractory lifetime | heats | ~4000 | ~4000 (2000 **) | 500–1500 | 500–1500 | ~1000 |
| Energy Output/Heat losses | % | 20–30 | 20–25 | 35–40 *** | 40–45 | 45–50 |
| Slag | % | 13 | 10 | 13 | 12 | 18 |
| offgas | % | 9 | 8 | 12–18 *** | 18–28 | 18–28 |
| Cooling water & others | % | 3 | 3 | 10 | 10 | 10 |

* End of treatment (before tapping), ** Hot bottom exchange, *** depending on scrap preheating, **** DR grade pellets, lower for BF grade DRI.

The EAF has a different slag practice from the BOF and is equipped with a slag door allowing slag overflow. The BOF is a closed furnace without slag loss during the blowing process and therefore has a higher yield and less heat loss. Typically, the yield on the BOF is slightly above 90%, whereas the EAF is below 90%. The lower refectory thickness and water-cooled side panel are other reasons for higher heat loss and energy output on the EAF.

## 6. Hybrid EAF–BOF Steelmaking

A scrap rate increase in BOF steelmaking has a good potential for optimization and reduction in $CO_2$ emissions on existing plants, which can be implemented quickly by keeping the current infrastructure. As discussed in Section 5, the BOF has advantages compared to the EAF in case of high hot metal ratios, but the solid charge mix is limited and, in addition to that, so is the potential for $CO_2$ emissions reduction. Hence, for the transition toward carbon neutrality, scrap-based EAF steelmaking and direction reduction-based steelmaking will also become more important. The integration of an electric arc furnace or an induction furnace (IF) for scrap premelting would be one of these options. Primetals

Technologies has developed this patent-pending process called the PREMELT process, where scrap and DRI are premelted with an electric arc furnace, mixed with the hot metal from the blast furnace, and then charged into the BOF converter; see Figure 13 [11]. The EAF focuses on melting only, whereas the main refining is completed in the BOF. The heat size of the EAF can be less than that of the BOF. That is why this solution is especially feasible for large BOF heats sizes of 300 t or more, where a direct replacement of the BOF with an EAF is challenging due to the large transformer size and lower productivity. An additional advantage of the PREMELT process is that the location of the EAF is very flexible and can be placed partially outside the steel plant, which does not require a change in internal steel plant logistics, and steel quality certification can remain unchanged. The furnace type and design for the PREMELT process depend on the charge mix and the heat size. For heat size lower than 70 t, IF is feasible, whereas for large heat sizes, which require more power input, EAF is more efficient. Concerning the charge mix, scrap is preferable for the REMELT process in the EAF, since recycled scrap has no additional $CO_2$ emissions and hence higher potential for overall $CO_2$ reduction. For a high scrap share of more than 90%, the highly efficient EAF Quantum is the best solution for this configuration due to the low energy consumption because off-gas is used for scrap preheating.

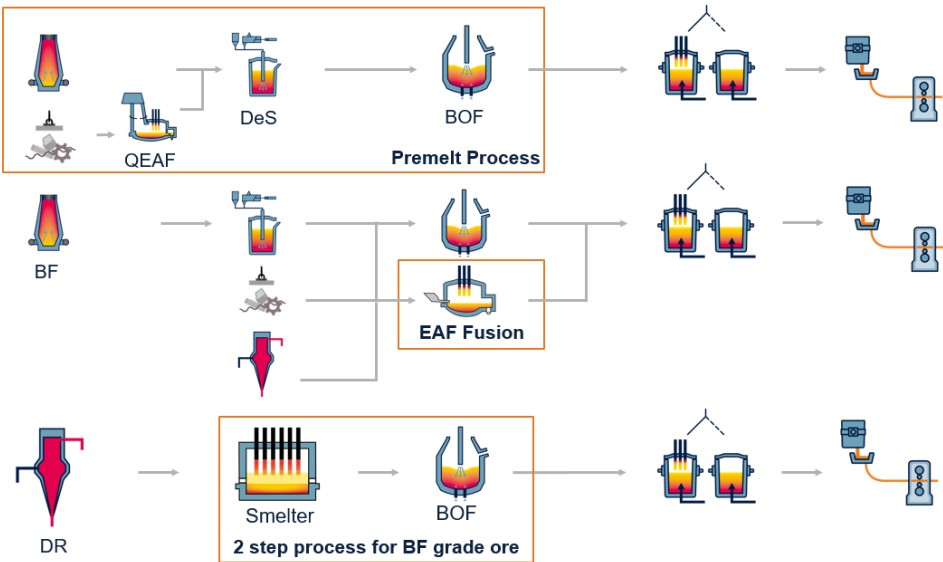

**Figure 13.** Hybrid EAF–BOF plants and possible routes for the transformation of BF grade ore based integrated steelmaking.

For plants that want to focus on scrap or high-grade DRI steelmaking and stop blast furnace operation in the long run, the direct replacement of BOF by EAF is one solution. The EAF with the same heat size as the BOF would do the melting and basic refining of the steel. In this case, the highest flexibility on the charge mix with hot metal, DRI, and scrap is required. The best furnace type for this configuration would be the EAF Fusion (see [10]). However, changes in layout, logistics, and steel quality certification will be required for such a configuration. In addition, very low limits of nitrogen content, for example, for exposed automotive grades, limit the operation of mainly scrap-based electric arc furnace steelmaking, whereas, with the BOF process and early application of argon for bottom stirring, nitrogen levels of less than 20 ppm are possible.

Close to zero $CO_2$ emissions will only be possible with fully scrap-based electric steelmaking powered by 100% green energy or in case of virgin materials by green hydrogen-based direct reduction. Since iron ore availability for the increasing share of direct reduction-based steelmaking will be limited, using lower-grade (BF grade) ore for DRI-based steelmaking will be necessary. For such raw material, a two-step process combining a smelter with a BOF is the better solution than direct processing in an EAF (see [12]).

## 7. Overview of $CO_2$ Reduction Potential for BOF Scrap Rate Increase

All measures described in the previous chapters allow for an increase in the scrap rate processed in the BOF converter and consequently a reduction in hot metal processed. An overview of the typical scrap rates and specific scope 1 and scope 2 $CO_2$ emissions of the BOF with different solutions applied can be seen in Figure 14 and more details to be found in the Supplementary Materials. The basis for this comparison is a standard BOF operation with a scrap rate of 16% and a non-optimized process using some cooling materials, limestone, raw dolomite, and others. With heat optimization (BOF and heat OPT), the scrap rate can be increased by 7% and up to 23%, and the resulting scope 1 direct emissions can be reduced mainly due to less carbon that needs to be oxidized. With DFPC lance, the scrap rate can be further increased, but $CO_2$ emissions from the BOF remain rather constant because additional oxygen is necessary, and more CO is combusted to $CO_2$. For heat optimization in combination with scrap preheating (SPH), the $CO_2$ emissions from the BOF as well as the combined BF–BOF emissions can be reduced, but the process time is extended. The additional energy for oxygen and natural gas for scrap preheating is included in the scope 2 emission but is relatively small. For coal injection in the case of the KOBM or JET process, BOF $CO_2$ emissions can increase compared to the base case, but combined BF and BOF $CO_2$ emissions can be reduced by more than 20%, which is shown by the orange line in the graph below. This graph shows a combination of heat optimization plus other solutions. It should be considered that these figures include plant-specific data and may vary for different steel plants.

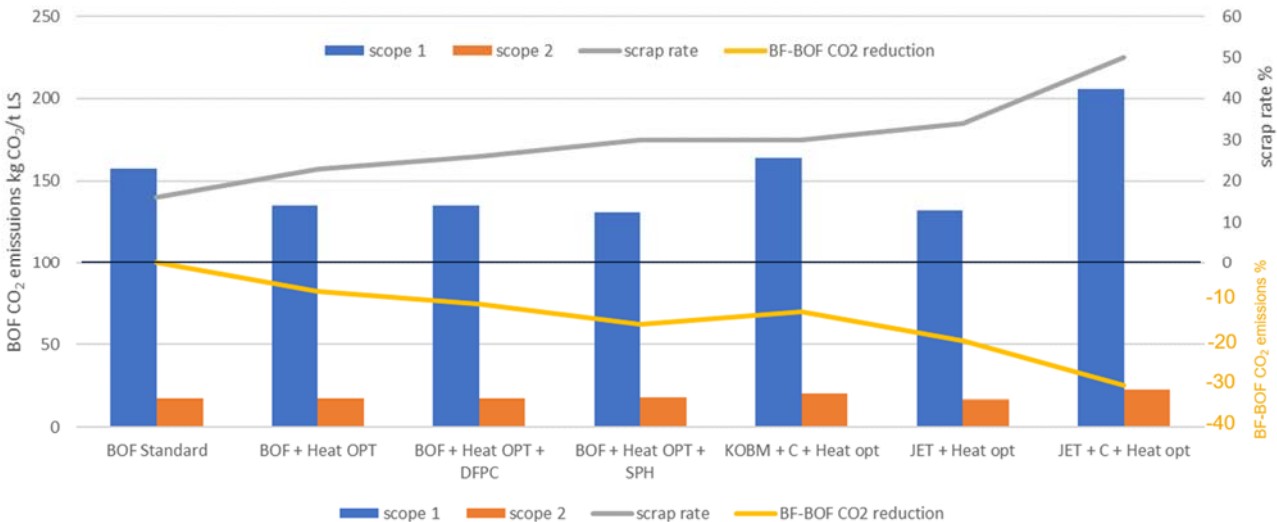

**Figure 14.** Specific $CO_2$ emissions (scope 1 and scope 2) per ton of liquid steel (LS) for different BOF solutions, including possible scrap rate and potential for reduction in total BF–BOF $CO_2$ emissions. Variation depending on operation conditions possible.

Since the BF has by far the largest amount of $CO_2$ emissions in the integrated steel-making route due to the coal-based ore reduction, replacing scrap with hot metal results in a similar amount of total BF–BOF $CO_2$ reduction. For example, with the dual flow post-combustion lance, the scrap rate can be increased by 4%, and a similar decrease in the total BF-BOF $CO_2$ emissions can be achieved. The results show that the higher the scrap rate, the higher the savings in $CO_2$ emissions compared to the base case. Even moderate additions of coal in the converter for heat balance, e.g., for the KOBM or JET converter, leads to savings in total $CO_2$ emissions in the end. These total savings occur because the savings for using less hot metal from the BF with specific emissions of 1500 kg $CO_2$ per ton of hot metal are much higher than the additional $CO_2$ emissions on the converter caused by the coal addition.

In Figure 15, the potential of every measure's BF–BOF $CO_2$ emissions reduction is compared with the complexity and the investment necessary for implementation. It can be seen, that upgrades with the scrap preheating lance, DFPC lance, and process automation for improved heat balance can be implemented rather simply, with a low investment budget and without changing the conventional BOF operation. KOBM converter upgrade does require a new bottom with a bottom blowing system and, in addition, the JET process requires space for a hot blast system. The most complex and highest investment from the solution discussed more in detail in this paper would be necessary for the PREMELT process where mainly scrap is melted externally in an EAF, mixed with the hot metal from the blast furnace, and then charged into the BOF. Replacing the BOF with a flexible EAF or changing to DRI-based steelmaking would have the highest potential for reducing the carbon footprint, but this would even exceed the complexity and investments necessary for the PREMELT process.

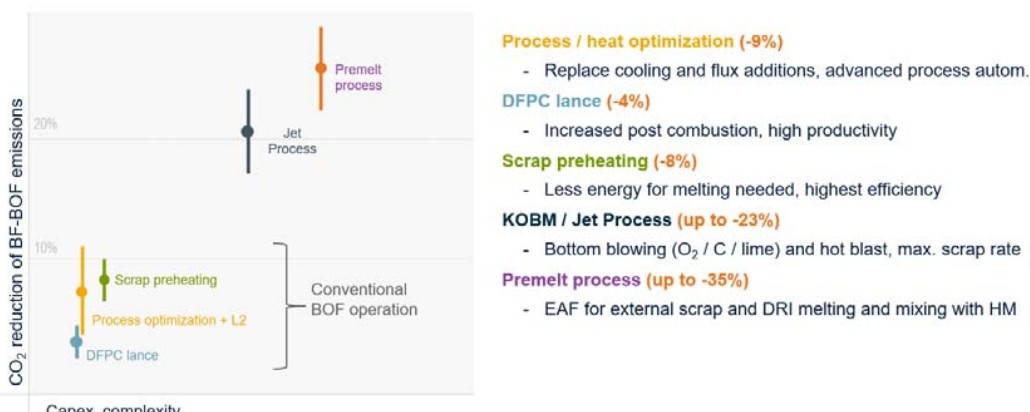

**Figure 15.** Investment complexity of solution for scrap rate increased in BOF steelmaking compared to its potential to reduce total BF–BOF carbon footprint.

## 8. Conclusions

The integrated BF–BOF steelmaking route is the most common steel production route, accounting for more than 70% of global steel production. Unfortunately, this route generates significantly higher $CO_2$ emissions than scrap-based and DRI-based electric steelmaking with an EAF. Transforming the integrated plants will be one of the major challenges in the upcoming years for the steel industry, and scrap as a zero-carbon recycling material will play an important role in this transformation process. The scrap availability is forecasted to grow significantly until 2050, especially in Asia. While only moderate growth is expected for global steel production, higher scrap rates in BOF steelmaking will support a significant decrease in the carbon footprint and increase in raw material flexibility in the steel industry. Several solutions for the increased scrap rate in the BOF already exist, such as simple upgrade packages for installation on existing converters such as process optimization, scrap preheating inside the converter, or a DFPC lance. Via revamping solutions that transform the existing BOF converter into a combined blowing converter or JET process, higher scrap rates of 30–50% can be achieved, and combined BF–BOF $CO_2$ emissions can be reduced by up to 30%. The integration of EAF in integrated steel plants allows a further increase in the solid charge rate but has the highest cost and complexity regarding implementation. For high scrap rates, residual elements and scrap quality will be limiting factors for many steel grades. However, design scrap and solutions for scrap sorting and processing are currently being developed to overcome this issue.

## 9. Patents

JET Patent portfolio, PREMELT process patent application.

**Supplementary Materials:** The following are available online at https://primetalstechnologies. clickmeeting.com/955251357/register.

**Author Contributions:** Conceptualization, B.V. and G.W.; methodology, B.V.; validation, E.W., K.P., G.W. and U.D.S.; data curation, B.V.; writing—original draft preparation, B.V.; writing—review and editing, U.D.S., E.W. and A.F.; visualization, K.P.; supervision, G.W. and A.F. All authors have read and agreed to the published version of the manuscript.

**Funding:** This research received no external funding.

**Informed Consent Statement:** Not applicable.

**Data Availability Statement:** Data sharing is not applicable.

**Conflicts of Interest:** The authors declare no conflict of interest.

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
