# Peer review of "Green LD (BOF) Steelmaking—Reduced CO2 Emissions via Increased Scrap Rate"

_metals, doi:10.3390/met12030466_

Round 1

Reviewer 1 Report

The paper presents green LD (BOF) steelmaking to reduce CO2 emission by increasing the scrap rate. It is of great importance to research green iron and steel making. However, this manuscript is like a review rather than an article. And it needs to be improved before it could be published.

  1. Scrap is commonly used in the EAF steelmaking process. What are the advantages of BOF steelmaking with a higher scrap rate compared to the EAF route?
  2. Solutions for increased scrap rate for BOF Steelmaking were discussed in this manuscript. The energy consumption and CO2 emission of those pretreatments and measures should be accounted into the whole process. So how to say there is zero CO2 emission with scrap as the raw material?
  3. There are some mistakes and typo errors in the manuscript. Please check and correct.

Author Response

Thanks for your review and the comments.

ad point 1: What are the advantages of BOF steelmaking with a higher scrap rate compared to the EAF route?

Update done, new Chapter 5 with comparison of BOF vs. EAF form typical reference data added, comparison table made. Of course optimum route is always depending on raw materials used and the steel grade targets. Normaly EAF and BOF have different scrap rates, charge mix --> direct comparison is not possible. Over 60% of Hot metal used the BOF is the more efficient furnace. For steel grades with very low nitrogen levels, such as external automotive grades, scrap based EAF steelmaking will be not feasible since due to the nitrogen pick up around open electric arc. For EAFs running with high DRI/HBI share lower Nitrogen levels are possible but still values are higher than for BOF. Energy output and heat losses on EAF are much higher than on BOF.

ad 2: Solutions for increased scrap rate for BOF Steelmaking were discussed in this manuscript. The energy consumption and CO2 emission of those pretreatments and measures should be accounted into the whole process. So how to say there is zero CO2 emission with scrap as the raw material?

In chapter 7 overview of CO2 reduction potential of overall BF-BOF emission is given. In figure 14 the energy consumption and additional CO2 emission from the measures like scrap preheating on the BOF is shown and in parallel also the potential for overall BF-BOF CO2 emission reduction is shown. Close to zero emissions will only be possible with fully scrap based electric steelmaking powered by 100% green electricity. The most promising technology for lowest CO2 emission based virgin material is from todays perspective hydrogen based direct reduction. (sentence added in chapter 6).

ad 3. There are some mistakes and typo errors in the manuscript. Please check and correct.

Update done

attached please find revised paper

Reviewer 2 Report

Paper deals with important topic and provides a good update on the trends in steelmaking related to increased scrap rate. However, there are several remarks.

Grammar.

Almost every sentence has grammar/syntax mistakes. Just very few examples are provided below (the entire paper shall be rewritten for language):

  • Line 10 “steelmaking will the most important challenge”
  • Line 11 – what “where” refers to
  • Line 12 refers “Several solutions” whereas Line 14 reads “solution includes“
  • Line 32 “ pledges…. covers”
  • Line 47 ”can be recycling“
  • Line 54 “Forecast are predicting”

and so on

Content

  • Line 11 – scrap also contains carbon
  • Line 32 - information is incorrect – steel sector is number one, according to IEA
  • Line 34 – reference is missing
  • Line 45 and further in the text - Meaning of scopes shall be explained
  • Line 71 “residual” and “trace” seem to be treated as synonyms but they not always are.
  • Paragraph starting line 160: why effect of carbon content of HBI/DRI (usually higher than that of steel scrap) is not taken into account?
  • 1 It seems that data for HM doesn’t include cokemaking and agglomeration (compare with Fig 5). Perhaps, it is reasonable to include those emissions as well.
  • GBFS abbreviation shall be explained
  • Section 4.1 – 1st paragraph sounds contradictory: if the process is optimized and the use of coolants becomes less necessary, it shall rather limit the use of scrap (which is also a coolant), not maximize it.
  • Line 214 – what is HMI?
  • Line 218 – It is not clear what is meant by “this technology”. From the content it looks like authors mean increasing Si content or temperature of HM. Is this the case? The former (although generates extra heat) is associated with (i) increased coke rate and CO2 emissions in BF and (ii) higher slag yield in BOF. Was this taken into account?
  • Line 223 – effect of FeSi addition can be offset by upstream (ferroalloys sector) energy use and CO2 emissions as well as cost efficiency. Was this considered?
  • Fig 6 (b) represents an interface not a model. Generally, some discussion of what is seen in this interface is desirable.
  • Line 243 – what is PT?
  • Fig 7 Why the slope of the curve dramatically changes after 700C? Comments are needed.
  • Fig 8 – caption doesn’t correspond to the content
  • Line 335 “oxygen ignition” doesn’t make sense – it is ignition of CO.
  • Line 343 Abbreviation KOBM stands for Klockner Oxygen Blown Maxhutte, not combined blowing converter
  • Line 345 Meaning of “softer” blowing shall be explained for a multidisciplinary reader.
  • Fig 10 (a) is it indeed (Fe) in slag or (FeO)?
  • Fig 10 (b) – abbreviations shall be explained
  • Line 379 – Not “coal already dissolved” but carbon dissolved
  • Line 381 – What is HTE?
  • Line 402 – “lows” => “allows”
  • Section 5 the difference between EAF, EAF Quantum and EAF Fusion shall be explained in more details.
  • Fig 14 Abbreviations such as OPT (or opt?), SPH etc. shall be explained. +C probably means coal injection, but this might not be obvious for a  multidisciplinary reader.

Author Response

Thanks for your remarks below please find our answers and attached updated revised version of paper.

Grammar.

Almost every sentence has grammar/syntax mistakes. Just very few examples are provided below (the entire paper shall be rewritten for language):

  • Line 10 “steelmaking will the most important challenge”, changed
  • Line 11 – what “where” refers to, update done
  • Line 12 refers “Several solutions” whereas Line 14 reads “solution includes“, changed
  • Line 32 “ pledges…. covers”, update done
  • Line 47 ”can be recycling“changed
  • Line 54 “Forecast are predicting” changed

fully grammar check done

Content

  • Line 11 – scrap also contains carbon, changed
  • Line 32 - information is incorrect – steel sector is number one, according to IEA, changed
  • Line 34 – reference is missing, Info is partly from COP 2021. Sentence was changed and reference added
  • Line 45 and further in the text - Meaning of scopes shall be explained, updated
  • Line 71 “residual” and “trace” seem to be treated as synonyms but they not always are. updated
  • Paragraph starting line 160: why effect of carbon content of HBI/DRI (usually higher than that of steel scrap) is not taken into account? Yes it is true that HBI/DRI based on natural gas has slightly higher carbon content than scrap which could provides additional chemical energy for the BOF heat balance. On the other hand HBI and DRI contain some percentage of iron oxide that was not reduced in the reduction plant as well as some non-metallic fraction called gangue which requires more energy for melting -->HBI /DRI has a higher cooling effect than Scrap and higher CO2 emission. For future H2 based direct reduction DRI will have no carbon content.
  • 1 It seems that data for HM doesn’t include cokemaking and agglomeration (compare with Fig 5). Perhaps, it is reasonable to include those emissions as well. Figure 1 is specific CO2 emissions per ton of steel, Figure 5 is per ton of output product of each process step.
  • GBFS abbreviation shall be explained, updated
  • Section 4.1 – 1st paragraph sounds contradictory: if the process is optimized and the use of coolants becomes less necessary, it shall rather limit the use of scrap (which is also a coolant), not maximize it. In this context optimization is meant for max. scrap rate and scrap as the main coolant. Text updated
  • Line 214 – what is HMI? Human Machine interface, updated
  • Line 218 – It is not clear what is meant by “this technology”. From the content it looks like authors mean increasing Si content or temperature of HM. Is this the case? The former (although generates extra heat) is associated with (i) increased coke rate and CO2 emissions in BF and (ii) higher slag yield in BOF. Was this taken into account? The target is not to increase the hotmetal temperature or the Si content, the target is more to optimize the BOF process with changing HM conditions, max. scrap rate and avoid other cooling additions. Text updated.
  • Line 223 – effect of FeSi addition can be offset by upstream (ferroalloys sector) energy use and CO2 emissions as well as cost efficiency. Was this considered? Yes, FeSi additions (heating additions) mainly for process corrections and only very minor scrap rate. Additional cost and CO2 emissions will arise.
  • Fig 6 (b) represents an interface not a model. Generally, some discussion of what is seen in this interface is desirable. Update done
  • Line 243 – what is PT? PT =Primetals technologies and is deleted, Update done
  • Fig 7 Why the slope of the curve dramatically changes after 700C? Comments are needed. The increase in slope is due to the change on specific heat capacity due to solid phase transformation in Fe at 723°C
  • Fig 8 – caption doesn’t correspond to the content, Update done
  • Line 335 “oxygen ignition” doesn’t make sense – it is ignition of CO. update done
  • Line 343 Abbreviation KOBM stands for Klockner Oxygen Blown Maxhutte, not combined blowing converter, update done
  • Line 345 Meaning of “softer” blowing shall be explained for a multidisciplinary reader. Soft blowing means higher lance height or less flow or both and is mentioned before
  • Fig 10 (a) is it indeed (Fe) in slag or (FeO)? It is Fe total in slag, this means oxidic and metallic content, FeO only would be slightly lower. Text update done
  • Fig 10 (b) – abbreviations shall be explained, Update done
  • Line 379 – Not “coal already dissolved” but carbon dissolved, yes update done
  • Line 381 – What is HTE?, heat transfer efficiency, update done
  • Line 402 – “lows” => “allows”, update done
  • Section 5 the difference between EAF, EAF Quantum and EAF Fusion shall be explained in more details., additional chapter for comparison of EAF and BOF steelmaking added were different EAF types are explained
  • Fig 14 Abbreviations such as OPT (or opt?), SPH etc. shall be explained. +C probably means coal injection, but this might not be obvious for a  multidisciplinary reader. update done in text above the figure